# *LcWRKY17*, a WRKY Transcription Factor from *Litsea cubeba*, Effectively Promotes Monoterpene Synthesis

**DOI:** 10.3390/ijms24087210

**Published:** 2023-04-13

**Authors:** Jing Gao, Yicun Chen, Ming Gao, Liwen Wu, Yunxiao Zhao, Yangdong Wang

**Affiliations:** 1State Key Laboratory of Tree Genetics and Breeding, Chinese Academy of Forestry, Beijing 100091, China; 2Research Institute of Subtropical Forestry, Chinese Academy of Forestry, Hangzhou 311400, China

**Keywords:** *Litsea cubeba*, WRKY, terpenoid, subcellular location, overexpression

## Abstract

The *WRKY* gene family is one of the most significant transcription factor (TF) families in higher plants and participates in many secondary metabolic processes in plants. *Litsea cubeba* (Lour.) Person is an important woody oil plant that is high in terpenoids. However, no studies have been conducted to investigate the WRKY TFs that regulate the synthesis of terpene in *L. cubeba*. This paper provides a comprehensive genomic analysis of the *LcWRKYs*. In the *L. cubeba* genome, 64 *LcWRKY* genes were discovered. According to a comparative phylogenetic study with *Arabidopsis thaliana*, these *L. cubeba WRKYs* were divided into three groups. Some *LcWRKY* genes may have arisen from gene duplication, but the majority of *LcWRKY* evolution has been driven by segmental duplication events. Based on transcriptome data, a consistent expression pattern of *LcWRKY17* and terpene synthase *LcTPS42* was found at different stages of *L. cubeba* fruit development. Furthermore, the function of *LcWRKY17* was verified by subcellular localization and transient overexpression, and overexpression of *LcWRKY17* promotes monoterpene synthesis. Meanwhile, dual-Luciferase and yeast one-hybrid (Y1H) experiments showed that the *LcWRKY17* transcription factor binds to W-box motifs of *LcTPS42* and enhances its transcription. In conclusion, this research provided a fundamental framework for future functional analysis of the *WRKY* gene families, as well as breeding improvement and the regulation of secondary metabolism in *L. cubeba*.

## 1. Introduction

Secondary metabolites from plants are a one-of-a-kind source of industrially significant biochemicals, flavors, and medicines [1,2]. Terpenoids are an essential secondary metabolite in plants and are commonly used as chemicals in dyes, flavors, fragrances, insecticides, and drugs by humans thanks to their significant economic worth [3,4]. To control the expression of genes in plants, transcription factors (TFs) coordinate a series of intricate regulatory networks [3,5]. Because there are many genes encoding different enzymes in the metabolic pathways of plants, manipulating TFs may be easier and more efficient than controlling the expression of the specific enzyme [6,7,8]. Numerous TFs have been demonstrated to be engaged in a number of physiological procedures in higher plants since the first TF was found in corn [9].

One of the major families of TFs in higher plants is the *WRKY* family [10]. Members of this family all contain two extremely preserved structural domains, a WRKYGQK sequence and a zinc finger structure [11,12,13]. The classification of WRKY proteins into three distinct groups is based on variations in the number and arrangement of structural domains, as well as the composition of their zinc-finger motif. Two WRKY domains and a zinc finger structure of the C2H2-type can be found in Group I proteins. One WRKY domain is present in both Group II and Group III proteins; however, Group II has a C2H2 zinc finger structure, whereas Group III has a C2HC zinc finger structure [12,14,15]. The WRKY protein binds the promoter elements of key enzymes in plants to regulate biological processes. The sweet potato gene *SPF1* was the first *WRKY* gene to be cloned, followed by *Arabidopsis thaliana* (72), *Oryza sativa* (103), *Solanum lycopersicum* (81), and *Zea mays* (120) [16,17,18]. WRKY plays a vital role in maintaining a range of physiological and biochemical responses in plants [19,20,21]. According to reports, WRKY TFs are engaged in many terpenoids’ biosynthesis, and most of the WRKY TFs regulating terpenoid synthesis are concentrated in Group I [22,23,24,25]. In a previous study, Xu et al. cloned *GaWRKY1*, the first WRKY TF involved in terpene synthesis from *Gossypium arboretum*, and showed that *GaWRKY1* can bind to the W-box of the CADI-A promoter [26]. It has been demonstrated that *AtWRKY44* in *A. thaliana* inhibits the synthesis of tannins in the seed coat [27,28]. *NaWRKY3* and *NaWRKY6* were connected to volatile terpene products in *Nicotiana tabacum* L. [29]. The terpenoid indole alkaloids (TIAs) are important anti-tumor substances and, in *Catharanthus roseus* L., *CrWRKY1* can influence TIAs by regulating tryptophan decarboxylase (TDC) gene expression [30].

*Litsea cubeba* (Lour.) Person is a member of the *Lauraceae* family and is found in Guangxi, Anhui, Zhejiang, and other provinces south of the Yangtze River in China. This plant is a valuable resource for spice production and holds great promise as a woody oil plant with significant potential for further development and application [31]. Essential oils are present in the fruits, flowers, and leaves of *L. cubeba*; citral is the primary constituent, which is widely used in cosmetics, soaps, and Chinese medicine thanks to its antibacterial properties. More than 90% of the contents of essential oil in *L. cubeba* are monoterpenes, such as citral, eucalyptol, pinene, and linalool. *L. cubeba* essential oil is primarily obtained from natural sources in the wild. As demand for natural oils increases, production must be increased immediately. However, the *WRKY* genes, which play a significant part in the production of plant terpenoids, have not been identified in *L. cubeba*. In this research, we collated whole genome data from *L. cubeba* and used bioinformatics to investigate the members of the WRKY TF family of *L. cubeba*. In particular, we concentrated on physical features, phylogenetic evolution, gene structure, structural motifs, cis-acting elements, and gene expression analysis. Furthermore, we validated the regulatory mechanism of target WRKY genes on terpenoid synthesis in *L. cubeba* by dual-LUC, Y1H, and overexpression experiments. This research lays a foundation for future analysis relating to *WRKY* regulation of plant terpenoid synthesis.

## 2. Results

### 2.1. Identification and Chromosomal Location of LcWRKYs

HMMER searches initially yielded 77 candidate genes. We eventually discovered 64 *WRKY* genes in the *L. cubeba* genome after manually removing the redundant genes (Appendix A). These were named *LcWRKY1*–*LcWRKY64* according to their position on the chromosome. These genes ranged from 124 amino acids to 1029 amino acids in length; these differences in gene length led to key differences in their expression.

The distribution of *LcWRKY* genes on the 12 chromosomes of *L. cubeba* was determined by chromosomal localization analysis (Appendix A). We can find these 64 *LcWRKY* genes scattered on the 12 chromosomes, with chromosome 4 having the most (14 genes), followed by chromosome 2 (nine genes), and chromosome 11 having the fewest (one gene). Six pairs of genes were also found to be closely linked in terms of their chromosomal position.

### 2.2. Motif Analysis and Structural Analysis of LcWRKYs

By analyzing the gene structure of *LcWRKYs*, it was evident that the protein-coding regions (CDS) of the *WRKY* gene range from 2 to 16 and that eight gene family members have no non-coding region (UTR). To research the structural characteristics of the evolutionary relationships among members of the *LcWRKYs* in *L. cubeba*, we utilized MEME to analyze the protein sequences of *LcWRKYs* (Figure 1). Motif 1 and motif 3 are conserved sequences of the WRKYGQK heptapeptide motif, and motif 2 is a zinc finger structural motif that is contained in all *LcWRKYs*. Motifs 1, 2, and 3 are the hallmark conserved structures of *LcWRKYs* transcription factors. Analysis showed that *LcWRKYs* contained 4 to 12 motifs; these motifs were essentially the same, thus indicating that these exert similar biological functions. Some of these proteins contain unique motifs that may be linked to unique functionality. These structural changes imply that *LcWRKYs* have undergone significant changes during their evolution compared with the basic structure of *WRKYs*.

### 2.3. Gene Duplication Analysis

A tandem repeat event is described as two or more genes located within 200 kb of a chromosomal region [32,33]. Our *LcWRKY* genes are clustered on *L. cubeba* chromosomes 2, 4, 5, 9, and 10 to form six tandem repeat event regions (*Lcu02G_04000/04001*, *Lcu04G_ 11429/11430*, *Lcu04G_13190/13191*, *Lcu05G_16661/16662*, *Lcu09G_24736/24740*, and *Lcu10G_26179/26191*) (Appendix A). Except for the above tandem repeat events, we identified 23 segmental repeat events by BLAST and MCScanX methods, which contained 39 *WRKY* genes (Appendix A). These findings imply that some *LcWRKY* genes may have developed by gene duplication and that segmental duplication events are the primary determinants of *LcWRKY* evolution.

### 2.4. Phylogenetic Analysis of WRKY Phylogeny in L. cubeba

To further investigate the phylogenetic status of *LcWRKYs*, we constructed a phylogenetic tree by multiple alignment for the protein sequences of *LcWRKYs* and *AtWRKYs*. We can see that *LcWRKY* proteins are clearly divided into three major groups. Group I featured 17 proteins, while Group III featured 7 proteins; these proteins may be involved in terpene synthesis (Appendix A). The largest group was Group II, which was further divided into five subgroups, five belonging to II-a, eight to II-b, ten to II-c, nine to II-d, and eight to II-e (Figure 2).

Combined with the published transcriptomic data (PRJNA763042, https://doi.org/10.1016/j.indcrop.2021.114423, accessed on 15 September 2022), we also analyzed the expression profiles of 64 *WRKY* genes at different developmental stages of the fruit (Figure 2). We found that these genes were specifically expressed in different developmental stages of the fruit and that the expression of genes was similar for genes clustered in the same group. Group I showed higher levels of expression during the later stages of fruit development, while Group III showed higher expression levels during both early and late fruit development. Some genes in Group II-d showed higher expression levels throughout the period of fruit development.

### 2.5. LcWRKY17, a Group I Protein, Was Co-Expressed with Genes Responsible for Terpene Synthesis

In a previous study, Group I *WRKY* genes were found to play an essential role in the control of terpenoid synthesis [34]. Weighted correlation network analysis (WGCNA) analysis showed that the 7 nodes genes of the terpenoid synthesis pathway showed extremely high connections with the other 69 edge genes. *LcWRKY17* was considered as the hub gene because of its association with more terpene-synthesis-related genes (Appendix A). Predicting the promoter cis-acting elements of terpene synthases (TPS) in *L. cubeba*, we found that these TPS promoters all contain WRKY binding element W-boxes ranging from 1 to 3 in number (Figure 3A). The transcriptome data were combined to further construct a clustering heat map of the expression of hub genes and the *TPS42* gene, a key enzyme for terpene synthesis. We could clearly see that *LcWRKY17* clustered with *TPS42* in the same family and that the expression profiles were correlated (Figure 3B). Analysis suggested that *LcWRKY17* may regulate the synthesis of terpenoids in *L. cubeba*.

### 2.6. Subcellular Localization of LcWRKY17

TFs are usually expressed and exert functionality in the nucleus. The full-length CDS of the cloned *LcWRKY17* gene was inserted into the pNC vector following removal of the stop codon (Appendix A), thus enabling these genes to fuse with GFP protein driven by the 35S promoter when expressed in *N. benthamiana* leaves (Figure 4A). The GFP protein in *N. benthamiana* (without gene insertion) was expressed throughout the entire cell; however, GFP protein fused with *LcWRKY17* proteins was only expressed in the nucleus, thus demonstrating that *LcWRKY17* genes are expressed and exert functionality in the nucleus (Figure 4B,C).

### 2.7. Quantitative Real-Time PCR (qRT-PCR)

The expression pattern of a gene is closely linked to its biological function. To further investigate the related functions of *LcWRKY17* and *TPS42*, we sampled *L. cubeba* fruits during different developmental periods and analyzed their expression profiles using qRT-PCR (Figure 3C). The results showed that *LcWRKY17* and *TPS42* have consistent expression profiles and showed peak expression levels at DAF60 and DAF120, thus corresponding to the critical period of essential oil synthesis [35,36]. These findings show that *LcWRKY17* may be crucial in the synthesis of fruit terpenoids.

### 2.8. Transient Overexpression of LcWRKY17 in L. cubeba

In order to examine the function of *LcWRKY17*, we adopted a simple and efficient transient expression approach because the stable transformation of *L. cubeba* is complex and time-consuming. The empty vector (*pNC-Cam2304-35S*) and the recombinant vector containing *LcWRKY17* (*pNC-Cam2304-35S-LcWRKY17*) were infiltrated by manual evacuation into the sterile seedling leaves of *L. cubeba* and analyzed for volatility after 72 h of growth [37]. The expression of *LcWRKY17* in *L. cubeba* leaves increased 4.3-fold and *LcTPS42* expression increased 4.1-fold compared with the control following transient expression (Figure 5A,B). The monoterpene content in the leaves of *L. cubeba* increased significantly when compared with the controls (Figure 5C). Transient overexpression of *LcWRKY17* enhanced the accumulation of major monoterpenes in *L. cubeba* leaves, such as α-pinene, camphene, β-myrcene, α-phellandrene, linalool, citronellal, neral, and geranial (Figure 5D). It has been shown that *LcTPS42* is highly expressed in the mid to late stages of *L. cubeba* fruit development and catalyzes the biosynthesis of the major monoterpene components geranium and linalool [38]. In this study, *LcWRKY17* transient expression was followed by a significant increase in *LcTPS42* expression and an increase in the content of major monoterpene components catalyzed by *LcTPS42*, indicating that WRKY transcription factors may catalyze the production of major monoterpene components through activation of *LcTPS42*.

### 2.9. LcWRKY17 Functions by Binding to the LcTPS42 Promoter Binding Element

*LcWRKY17* is a WRKY TF member, which bound to the W-box element in the promoter of terpene synthase genes to regulate plant secondary metabolic synthesis. Using dual-LUC assay, we determined the regulatory effect of *LcWRKY17* transcription factor on the *LcTPS42* promoter. When compared with the control null, the dual-LUC activity experiment showed that the WRKY transcription factor *LcWRKY17* significantly activated the *LcTPS42* promoter (Figure 6A).

We used the Y1H assay to advance our understanding of the *LcWRKY17* transcription factor’s regulatory mechanism on the *LcTPS42* promoter. Y1H assay revealed that co-transformation of three tandem repeats of the W-box element and pGADT7-*LcWRKY17* vectors successfully activated AbA gene expression and enabled yeast to grow on Aureobasidin A (AbA) antibiotic plates, while mutant W-box (m W-box) element and pGADT7-*LcWRKY17* co-transformation could not grow on plates containing AbA antibiotics (Figure 6B). The results suggested that *LcWRKY17* binds to the W-box motif of the *LcTPS42* promoter. Thus, *LcWRKY17* regulates the synthesis of monoterpenoids by binding to the W-box elements of the *LcTPS42* promoter.

## 3. Discussion

In our research, we discovered 64 *WRKY* genes in *L. cubeba* and identified a significantly higher monoterpene content in *L. cubeba* when we overexpressed *LcWRKY17,* a member of Group I. Our research shows that WRKY TFs are crucial for *L. cubeba*’s terpene production.

As more and more whole genomes are released, identifying and analyzing TFs at the whole genome level has become an important focus of genomics research. As the most essential TF in plants, WRKY has been identified in tobacco, rice, cucumber, grapevine, and poplar [13,18,39,40,41]. However, no similar research has been conducted in *L. cubeba* and the specific function of the *LcWRKY* gene remains unclear [28]. After the *L. cubeba* genome assembly and sequencing were finished, *LcWRKYs* can be identified and analyzed more efficiently. In this research, we identified and analyzed the *LcWRKYs* for the first time. The first step in studying the functionality of gene families is to classify them. A classification system for the *WRKY* gene family was previously developed for *Arabidopsis* that is now widely accepted. *WRKY* genes in plants are divided into three main groups, and members of the Group Ⅱ can be further separated into five subfamilies because they are not monophyletic [15,16]. Meanwhile, the variety of zinc finger structures can be linked to the diversity of WRKYs [42]. We discovered 64 *LcWRKY* genes in the *L. cubeba* genome database. The proteins of *LcWRKYs* and *AtWRKYs* were then used to create a phylogenetic tree. Group II was the one containing the most *WRKY* genes, accounting for 62.5% of the total. This phenomenon was similarly demonstrated in *Arabidopsis*, *Brassica napus*, and *Brassica rapa* [43].

Tandem duplication and segmental duplication produce distinct evolutionary processes for duplicated genes [44]. It is generally considered that duplicated genes located within a 200 Kb region on the same chromosome arise from tandem repeats, while duplicated genes on different chromosomes arise from segmental repeats [45,46]. We identified 12 tandem duplication genes and 23 segmental duplication events in *L. cubeba,* demonstrating the importance of segmental duplication in the expansion of the *LcWRKYs*. To adapt to different development environments, these duplicated *LcWRKY* genes most likely created novel gene functions. Repetitive genes are crucial for plants to adapt to challenging and shifting surroundings [44,47]. Repeat genes may undergo sub-functionalization, new functionalization, and loss during continuous evolution [48]. It has been previously demonstrated that two genes on the same chromosome that are close to one another, particularly two tandem repeat genes, are more likely to be co-regulated.

Expression patterns are closely related to gene function [49]. It has been shown that *WRKY* genes play a crucial role in controlling secondary metabolism, plant growth and development, and plant responses to various abiotic stressors [50,51,52]. Analysis of the expression pattern of the *LcWRKY17* gene during different periods of fruit development in *L. cubeba* showed that *LcWRKY17* was strongly expressed during the fruit’s middle and late phases of growth. Furthermore, the expression profiles of *LcWRKY17* and *LcTPS42*, a key enzyme for monoterpene synthesis, were consistent, both being specifically highly expressed in the middle and late stages of fruit development. Combined with these results, we believe that *LcWRKY17* is essential for the terpene synthesis pathway.

WRKY TFs are crucial in the synthesis of plant secondary metabolites. Terpenoids are a significant part of plant secondary metabolites. Strategies for the synthesis of terpenoids in plants include the mevalonate (MVA) and mevalonate-independent (MEP) pathways [53,54]. These are controlled by a number of structural genes in the biosynthesis route, and TFs are responsible for secondary regulation [55,56]. The majority of artemisinin synthesis genes have been demonstrated to express themselves more frequently when *AaWRKY1* is present, suggesting that the *AaWRKY1* TF regulates the production of artemisinin [57,58]. *AaWRKY9* regulates artemisinin biosynthesis in *Artemisia annua* via the mediation of light and jasmonic acid [34]. In *S. lycopersicum, SlWRKY71* can regulate the expression of terpene synthase [59]. *SlWRKY35* activates the MEP pathway in tomato fruit, thereby promoting carotenoid biosynthesis [35]. In this study, the transient overexpression of *LcWRKY17* in *L. cubeba* significantly promoted monoterpenes in *L. cubeba*. Y1H and dual-LUC experiments further showed that *LcWRKY7* directly binds to the W-box of the *LcTPS42* promoter and activates its transcription, thereby promoting monoterpene synthesis [38].

## 4. Materials and Methods

### 4.1. Plant Materials and Treatment

*L. cubeba* samples were collected by our group and planted in the Fuyang field of Hangzhou, China (30°27′94″ N, 119°58′43″ E). ‘Anhui 3’ was used as the seedling material for group culture in transient transformation experiments. After sterilization, adventitious shoots were induced in the succession medium for about 20–30 days. The adventitious shoots were replaced with a new succession medium for proliferation. Seedlings of *Nicotiana benthamiana* were cultivated in a greenhouse.

### 4.2. Identification of WRKY Genes in L. cubeba

The Pfam number of the *WRKY* gene family (PF03106) was obtained through literature and the hidden Markov model (HMM) files of all gene families were downloaded from the Pfam Protein Family Database (http://bioinformatics.psb.ugent.be/webtools/plantcare/html/, accessed on 22 June 2022). Combined with the genomic data of *L. cubeba* obtained from our group’s previous research [38], all *WRKY* gene families in *L. cubeba* were screened and identified using the Simple HMM Search plug-in in TBtools (https://github.com/CJ-Chen/TBtools, accessed on 22 June 2022). Genes with an e-value < 10^−10^, along with duplicated genes, were removed to provide a final suite of target genes containing *WRKY* structural domains in *L. cubeba*.

### 4.3. Chromosomal Location Analysis

The *WRKY* gene’s location on the chromosome was determined using the *L. cubeba* genome’s chromosomal annotation file, and its distribution on the chromosome was mapped using MapGene2Chromosome V2 (http://mg2c.iask.in/mg2c_v2.0/, accessed on 24 June 2022).

### 4.4. Motif Analysis and Structural Gene Analysis

Protein sequences and CDS sequences of the *WRKY* genes were first retrieved by searching genomic protein files and CDS files. Protein sequences of the genes in the *WRKY* gene family were analyzed using MEME (https://memesuite.org/meme/tools/meme, accessed on 8 July 2022). The *WRKY* gene family’s structures were examined using the Gene Structure View plug-in of the TBtools (https://github.com/CJ-Chen/TBtools, accessed on 8 July 2022), with a threshold of 30 motifs.

### 4.5. Gene Duplication Analysis

Gene duplication occurrences were examined using MCScanX [60]. According to geographic information from the *L. cubeba* genomic database, all *LcWRKY* genes were assigned to the *L. cubeba* chromosome. The results were visualized using the Advanced Circos plugin in TBtools (https://github.com/CJ-Chen/TBtools, accessed on 24 July 2022).

### 4.6. Evolutionary Analysis of the WRKY Gene Family

MEGA7.0 was used to perform protein sequence alignments of the *LcWRKY* and *AtWRKY* genes, and the alignment files were produced in phylip3.0 format [61]. The maximum likelihood (ML) approach of RAxML was used to build phylogenetic trees on the internet site CIPRES (https://www.phylo.org/portal2/login!input.action, accessed on 15 July 2022).

### 4.7. Expression Analysis

The expression patterns of the fruit development of *L. cubeba* were analyzed based on published RNA-seq data (PRJNA763042) by the subject group (https://doi.org/10.1016/j.indcrop.2021.114423, accessed on 15 September 2022) [37]. Using the FPKM (fragments per kilobase million) value of 38,988 differentially expressed transcription factors (DETs) at the 12 stages of fruit development of *L. cubeba*, we employed the WGCNA approach of the R package with a weighted cut-off value >0.50 [62,63]. The Kyoto Encyclopedia of Genes and Genomes (KEGG) database was used to obtain keywords and pathways (http://www.kegg.jp/blastkoala/, accessed on 15 September 2022). We selected genes related to terpene synthesis and visualized these with Cytoscape.

### 4.8. Cis-Acting Element Analysis

TBtools program was used to extract the promoter sequences of *L. cubeba* WRKY family members, detected and identified using the online Plant CARE website (http://bioinformatics.psb.ugent.be/webtools/plantcare/html/, accessed on 22 September 2022) for cis-element detection. The outcomes were then displayed using tBtools.

### 4.9. LcWRKY17 Gene Cloning

*LcWRKY17* was amplified using cDNA from different tissues of *L. cubeba*. Primer 3 (https://bioinfo.ut.ee/primer3-0.4.0/, accessed on 15 September 2022) was responsible for creating the cloning primers. High-fidelity enzymes were selected for amplification (MCLAB, Beijing, China); the amplification system and conditions were in accordance with the manufacturer’s instructions. A Gel Extraction Kit (OMEGA, Beijing, China) was used to purify the PCR products and ligate them to the vector for further sequencing (Appendix A).

### 4.10. Subcellular Localization Analysis

Subcellular localization experiments were utilized to investigate the sites of gene expression and action. The correctly amplified and sequenced complete gene (following removal of the stop codon) was introduced into the *pNC-Green-SubC* plant expression vector and transferred into *Agrobacterium* GV3101. *Agrobacterium* containing the target vector was then transferred into leaves of *N. benthamiana* cultivated for approximately four weeks. The results were observed by Echo Revolve fluorescence microscopy (Revolve FL) after two days, with a red nuclear localization marker as a positive control.

### 4.11. Quantitative Real-Time PCR (qRT-PCR)

RN38 EASY Spin Plus Plant kit (Aidlab, Beijing, China) was used to extract total RNA. An ABI PRISM 7500 instrument and a TB Green^®^ Premix Ex TaqTM II kit (TaKaRa, Tokyo, Japan) were used to perform qRT-PCR. Using the *UBC* gene as an internal control [38]. Primer Premier 3.0 was used to create primers for qRT-PCR reactions; the primer sequences are provided in Appendix A. The relative expression levels were calculated by the 2^−ΔΔCT^ approach [37].

### 4.12. Transient Overexpression of LcWRKY17 in L. cubeba

To investigate the role of *LcWRKY17* overexpression on terpenoid synthesis in *L. cubeba*, we performed *LcWRKY17* overexpression experiments on sterile *L. cubeba* seedlings [37]. The empty vector (*pNC-Cam2304-35S*) and the recombinant vector containing *LcWRKY17* (*pNC-Cam2304-35S-LcWRKY17*) were each transformed into *Agrobacterium* LBA4404 strains and infiltrated by manual evacuation into the leaves of seedlings of *L. cubeba* with similar growth rates. Detection after culture at 26 °C for 50–72 h.

### 4.13. Dual-Luciferase and Yeast One-Hybrid (Y1H) Assays

Dual-LUC assay was used to detect the regulatory effect of the *LcWRKY17* transcription factor on the *LcTPS42* promoter. The *LcTPS42* promoter (about 2000 bp upstream of the start codon) was constructed into the pGreenII0800-LUC vector and the *LcWRKY17* gene was constructed into the pGreenII62-SK vector. Agrobacterium strain GV3101 (with pSoup) was individually transformed with the recombinant vector. Tobacco leaves were infested with a mixture of Agrobacterium cells. Using a dual LUC test kit (Promega, Madison, WI, USA), the LUC and REN enzyme activities were assessed after 2–3 days of incubation in the light incubator.

The *LcWRKY17* transcription factor’s ability to bind to the W-box elements of the *LcTPS42* promoter was examined using the Y1H assay. The *LcWRKY17* transcription factor was constructed into the pGADT7 vector. Three tandem repeats of the *LcTPS42* promoter binding element W-box were constructed into the pAbAi vector with BstBI enzyme digestion and gum recovery. The pGADT7 vector containing the target fragment and the pAbAi vector were simultaneously transformed into yeast for validation. The transformed yeast cells were grown on SD/-Leu/-Ura medium with and without Aureobasidin A (AbA), then the plates were kept at 30 °C for 2–3 days.

## 5. Conclusions

In our study, we performed genome-wide analysis of the *L. cubeba WRKY* gene family with a particular focus on their regulatory role on secondary metabolism. *L. cubeba* possessed 64 *WRKY* genes, which are distributed on 12 different chromosomes. Evolutionary analysis divided these 64 *WRKY* genes into three groups, and Group Ⅱ was further divided into five subgroups; this classification was clearly supported by their gene structures. Furthermore, transient overexpression experiments showed that *LcWRKY17* specifically regulated the synthesis of monoterpenoids in *L. cubeba*. Dual-LUC and Y1H analyses further revealed that *LcWRKY17* promotes monoterpene synthesis by binding to the W-box elements of the *LcTPS42* promoter and activating its transcription. Collectively, our research offers a theoretical foundation for continued in-depth investigation into the function of *LcWRKYs* and their regulatory role on plant terpenoids.

## Figures and Tables

**Figure 1 ijms-24-07210-f001:**
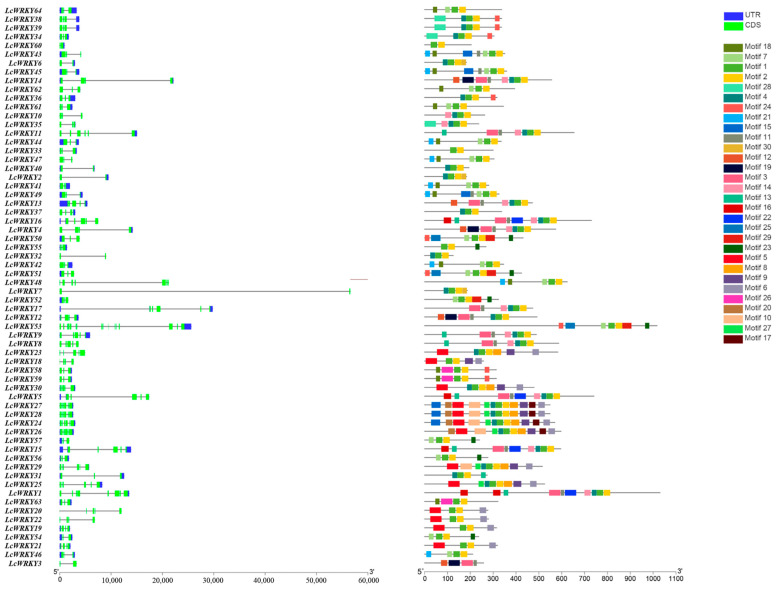
Gene structure and motif analysis of *LcWRKYs*. The blue boxes, green boxes, and thin black lines represent the UTR, CDS, and introns, respectively. MEME analysis revealed conserved motifs of *LcWRKY* proteins. Colored boxes on the right denote 30 motifs.

**Figure 2 ijms-24-07210-f002:**
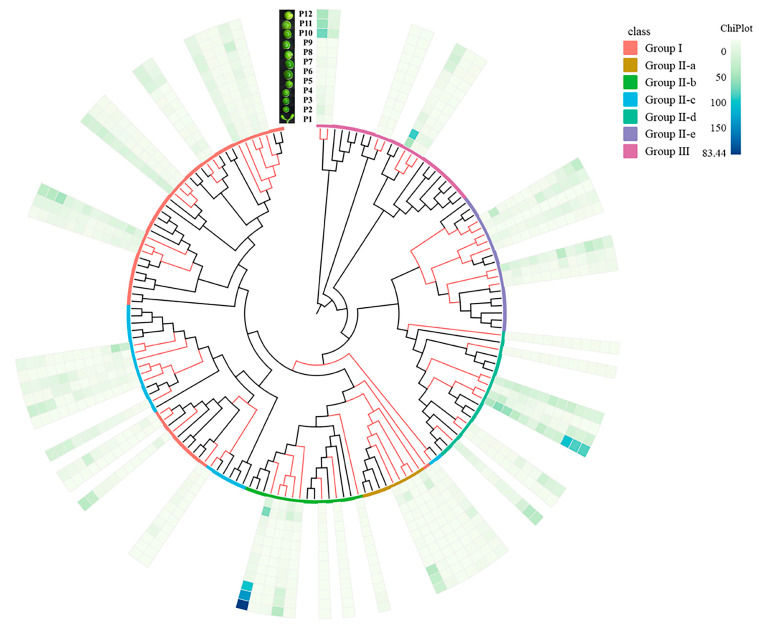
Maximum-likelihood phylogenetic tree of WRKY proteins in *L. cubeba* and *A. thaliana*. The phylogenetic analysis was constructed by MEGA11 software with bootstrap test of 1000 times. Seven subfamilies of WRKYs are distinguished by circles of different colors (Group I, Group IIa–e and Group III). Red and black branches represent *LcWRKYs* and *AtWRKYs*. The expression pattern of of *LcWRKYs in* different developmental periods of the *L. cubeba* fruits were investigated based on the RNA-seq data (PRJNA763042). P1–12 represent different developmental stages of fruits from *L. cubeba*: 15, 30, 45, 60, 75, 90, 105, 120, 135, 150 days after full bloom (DAF).

**Figure 3 ijms-24-07210-f003:**
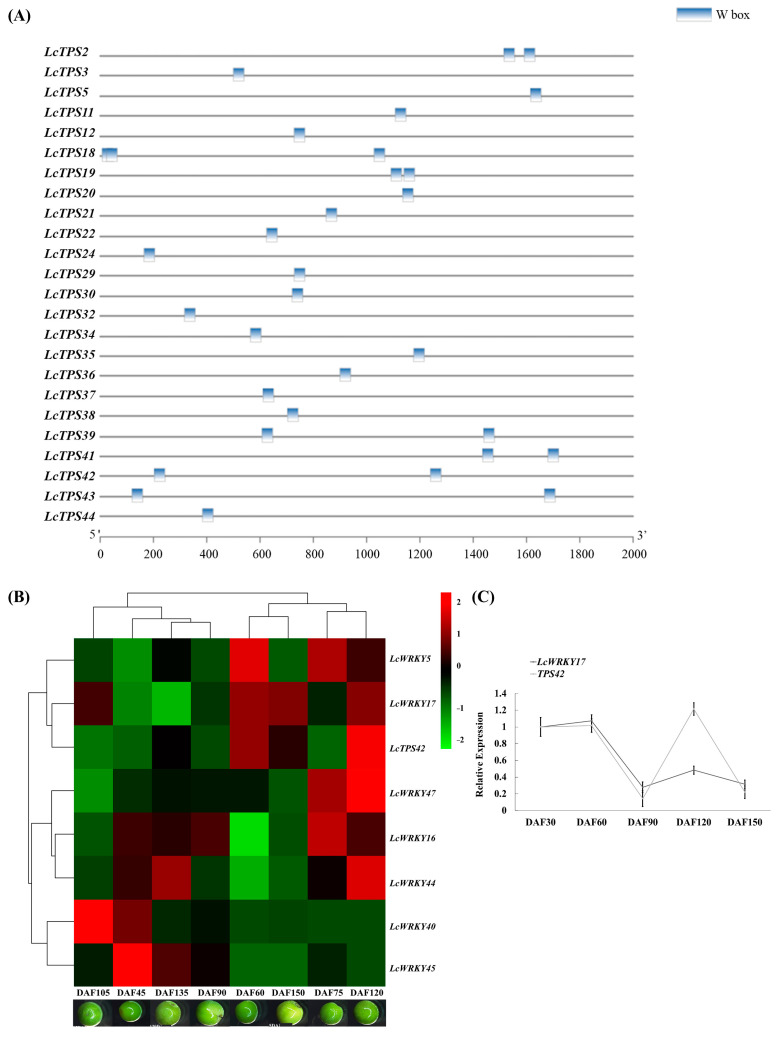
(**A**) Analysis of TPS (terpene synthases) promoter cis-acting elements in *L. cubeba*. (**B**) Cluster analysis of hub genes and TPS42. (**C**) Gene expression of *LcWRKY17* and *LcTPS42* in different developmental stages of the fruit (day after flower, DAF). UBC gene is the internal reference, the expression value of the first sample DAF30 is set to 1, and the data are represented as the mean. The error bars represent the standard deviation of three biological repeats. Based on Student’s *t*-test, asterisks indicate statistically significant differences from sample 1.

**Figure 4 ijms-24-07210-f004:**
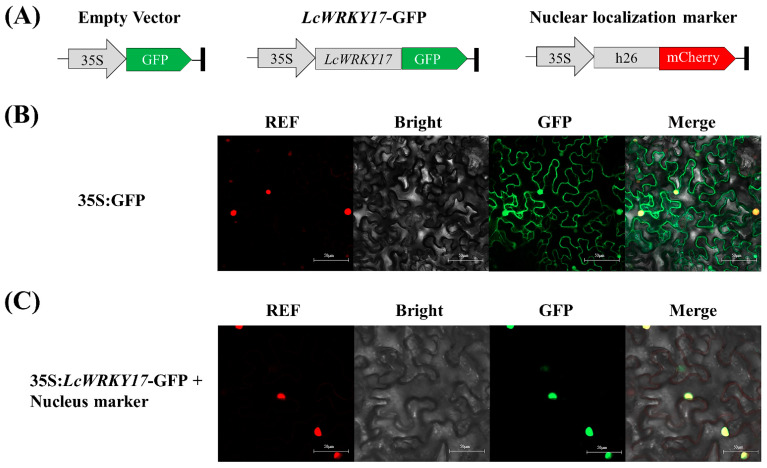
Subcellular localization of *LcWRKY17* in *N. benthamiana* leaves. (**A**) Schematic diagram of vector. (**B**) Empty vector (35S: GFP). (**C**) 35S: *LcWRKY17*-GFP and nucleus marker infected four-week large tobacco leaves. Pictures show REF, Bright, GFP, and Merge from left to right. Red represents nuclear localization marker, green represents green fluorescent signal, and yellow represents localization in the nucleus. Scare bar = 50 μm.

**Figure 5 ijms-24-07210-f005:**
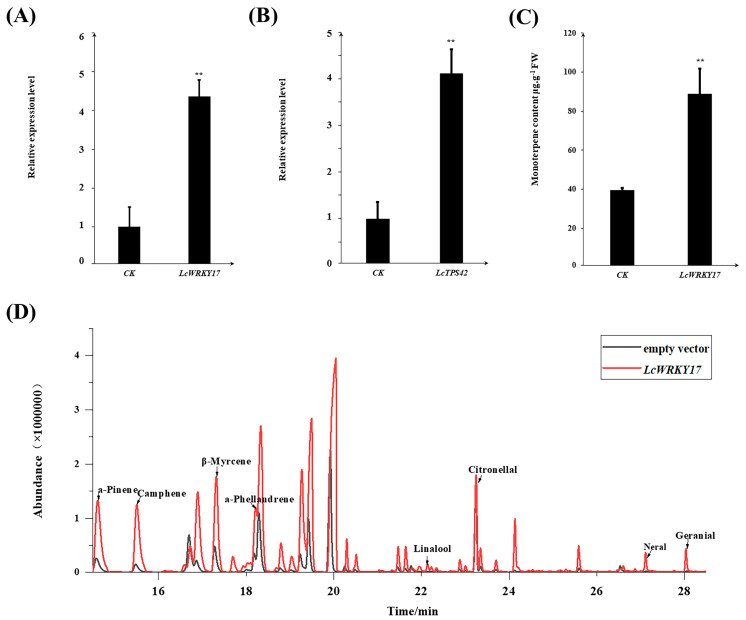
(**A**) Relative expression of *LcWRKY17* in transient overexpression in *L. cubeba*. (**B**) Relative expression of *TPS42* in transient overexpression in *L. cubeba*. (**C**) Monoterpene contents in leaves of *L. cubeba* after transient overexpression of *LcWRKY17*. (**D**) Volatile components in leaves of *L. cubeba* with transient overexpression of *LcWRKY17*. Data represent the mean ± SDs of three biological replicates. Student’s *t*-test was used to assess confidence levels (CK = blank control) (**, *p <* 0.01).

**Figure 6 ijms-24-07210-f006:**
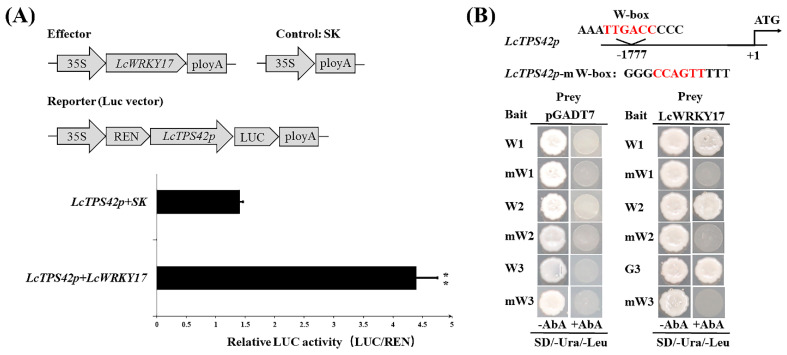
*LcWRKY17* binds directly to the *LcTPS42* promoter element. (**A**) Dual-LUC analysis showed that *LcWRKY17* activated the *LcTPS42* promoter (**, *p* < 0.01). (**B**) Y1H analysis showed that *LcWRKY17* binds to the W-box elements of the *LcTPS42* promoter. Red font indicates the sequence of W-box element and the sequence of mutant element.

## Data Availability

The datasets supporting the conclusions of this article are available in the NCBI Short Read Archive under accession number PRJNA763042. https://www.ncbi.nlm.nih.gov/bioproject/PRJNA763042, accessed on 15 September 2022.

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
