# Peer review of "LcWRKY17, a WRKY Transcription Factor from Litsea cubeba, Effectively Promotes Monoterpene Synthesis"

_ijms, 2023, doi:10.3390/ijms24087210_

Round 1
Reviewer 1 Report
All my comments are in the PDF. I didnt want to copy paste them again.
Very nice work - i will not summarize, BUT you need to fix the errors before submission:
Figure 1 should be in supplements and the phylogenetic tree should be Figure 1, which is mentioned in the methods and results but not shown and is not in the supplements. Just Lc alone would be fine, but against Athal or rice would be interesting.
*The figures' resolution (of the original I got, since my commented copy has an even lower resolution) is too low for all!!! Such data is only useful if we can zoom onto it.
The 5. CONCLUSIONS paragraph comes after the Methods? intentional?? Did i miss something about format?
The REFs were not finished properly, lots of {NUM} here and there. Someone was in a rush.
Abbreviations are used too often without explanation, all noted in PDF.
Fig3A is not convincing and confusing. I do not see that as evidence for anything. See comments.
Please see all my comments in the PDF. The paper is sound, but these corrections will make it top-notch.

Reviewer 2 Report
Line Nos- 41-42 Reframe the sentence “ Three groups of WRKY proteins can be distinguished depends on the quantity of structural domains and the design of their zinc-finger motif”.
Line Nos 47-48. “The sweet potato gene SPF1 was used to 47 clone the first WRKY gene {#9}- is SPF1 a WRLY gene itself? If not how WRKLY gene was cloned? Reframe the sentence.
Line No 49- Change “Vital part” into “vital role”
Line Nos 56-57 - NaWRKY3 and NaWRKY6 were connected to volatile terpene products in Nicotiana tabacum L., according to research by Kibbe et al [30]. Remove “according to research by Kibbe et al [30].” And write Kibbe et al 30 in Parenthsis i..e NaWRKY3 and NaWRKY6 were connected to volatile 56 terpene products in Nicotiana tabacum L.[ Kibbe et al (30)].
Line No 59 - Expand “TIA” and “ TDC” into full form
Line Nos 61-62: Reframe the sentence” This is one of the most important spice plant resources and a woody oil plant with excellent potential for development and application [32]”
Line no 68: “the supply still exceeds demand”, or is it “demand still exceeds supply?
Line No 69: Is the “ 56,568 amino acid” number correct?
Line No 119: Change the word “Expect” to “Except”.
Line Nos 128-129: “these proteins are involved in terpene synthesis”. How the authors could confirm without functional characterization? Authors may add a table of functionally characterized WRKY genes in other plants with references showing homologs with their WREKY genes.
Line No 114 : 2.3 and 4.5 (Materials and methods) Covariance analysis . Why the authors term the gene duplication analysis as covariance analysis? Why not the term “gene duplication analysis”?
Line nos 142-143: WGCNA analysis showed that the LcWRKY17 gene is a hub gene for terpenoids synthesis and plays a significant part in the synthesis of terpenoids (Fig. 3A).
The authors did not mention on what data they did this analysis and found the hub gene? Did authors analysis transcriptome data ? If so which data? This should be elaborated giving the source of data and information of the data.
The legends of Fig 2 and Fig 3 are same. Authors should check thoroughly before submitting the manuscript to the journal.
The legend of the Fig 3 should be self-explanatory and the details should be given in the materials and methods as well.
Round 2
Reviewer 1 Report
good job. the text and story is improved.
I was not really able to open the supp. file easily. (40MB for a Word file is too big).
the figure resolution was not improved by making a larger file size; but it is as it is. I recommend checking the journal requirements one more time.
But the general idea of having the images available as supplements is okay. Maybe save as a PDF or keep as images. But please ask the journal what they would like to have.
Reviewer 2 Report
Fig 1 is not cited. Fig.1 legend is not clear and self-explanatory.
Need details of transcriptome sequencing, assembly, and other analysis details in the materials methods and results section. The authors have not mentioned this in detail. Further, authors need to provide the Accession Nos of transcriptome data, deposited in the public database.
